# Aurivillius Oxides Nanosheets-Based Photocatalysts for Efficient Oxidation of Malachite Green Dye

**DOI:** 10.3390/ijms23105422

**Published:** 2022-05-12

**Authors:** David A. Collu, Cristina Carucci, Marco Piludu, Drew F. Parsons, Andrea Salis

**Affiliations:** 1Department of Chemical and Geological Sciences, University of Cagliari, S.P. 8 km 0.700, 09042 Monserrato, Italy; david_collu@outlook.it (D.A.C.); cristina.carucci@unica.it (C.C.); drew.parsons@unica.it (D.F.P.); 2Consorzio Interuniversitario per lo Sviluppo dei Sistemi a Grande Interfase (CSGI), Via della Lastruccia 3, 50019 Sesto Fiorentino, Italy; 3Department of Biomedical Sciences, University of Cagliari, S.P. 8 km 0.700, 09042 Monserrato, Italy; mpiludu@unica.it

**Keywords:** nanosheets, perovskites, photocatalysis, Aurivillius oxides, malachite green, organic dyes

## Abstract

Aurivillius oxides ferroelectric layered materials are formed by bismuth oxide and pseu-do-perovskite layers. They have a good ionic conductivity, which is beneficial for various photo-catalyzed reactions. Here, we synthesized ultra-thin nanosheets of two different Aurivillius oxides, Bi_2_WO_6_ (BWO) and Bi_2_MoO_6_ (BMO), by using a hard-template process. All materials were characterized through XRD, TEM, FTIR, TGA/DSC, DLS/ELS, DRS, UV-Vis. Band gap material (E_g_) and potential of the valence band (E_VB_) were calculated for BWO and BMO. In contrast to previous reports on the use of multi composite materials, a new procedure for photocatalytic efficient BMO nanosheets was developed. The procedure, with an additional step only, avoids the use of composite materials, improves crystal structure, and strongly reduces impurities. BWO and BMO were used as photocatalysts for the degradation of the water pollutant dye malachite green (MG). MG removal kinetics was fitted with Langmuir—Hinshelwood model obtaining a kinetic constant k = 7.81 × 10^−2^ min^−1^ for BWO and k = 9.27 × 10^−2^ min^−1^ for BMO. Photocatalytic dye degradation was highly effective, reaching 89% and 91% MG removal for BWO and BMO, respectively. A control experiment, carried out in the absence of light, allowed to quantify the contribution of adsorption to MG removal process. Adsorption contributed to MG removal by a 51% for BWO and only by a 19% for BMO, suggesting a different degradation mechanism for the two photocatalysts. The advanced MG degradation process due to BMO is likely caused by the high crystallinity of the material synthetized with the new procedure. Reuse tests demonstrated that both photocatalysts are highly active and stable reaching a MG removal up to 95% at the 10th reaction cycle. These results demonstrate that BMO nanosheets, synthesized with an easy additional step, achieved the best degradation performance, and can be successfully used for environmental remediation applications.

## 1. Introduction

Pollution is one of the main issues that afflict modern society. Pollutants include heavy metals, inorganic, and organic compounds [1,2,3,4]. Among organic compounds, pharmaceuticals, hormones, sweeteners, cleaning products, and dyes have been largely studied for their toxicity [5]. The first step to tackle pollution is to detect and quantify the specific contaminants [6]. Due to their toxicity, heavy metals traces are critical for natural environments, hence, they have to be precisely determined [7,8]. Organic dyes are one class of substances extensively used in the textile industry, which can also be found in a huge number of products, such as plastics, toys, cosmetics, and coatings [9,10,11]. Some organic dyes are toxic and carcinogenic, and can be found in textile industry wastewaters [12,13]. It is estimated that about 15% of the total dyes produced annually in the world ends up in the environment [12]. The main strategies to solve the problem of dye contamination involve their removal through the use of either adsorbents [14], electrochemical degradation [15,16], or photocatalysts [12]. Synthetic dyes are specifically designed to be very resistant to aerobic degradation, although they are not resistant to anaerobic conditions [17]. About 70% of the total organic dye pollution is composed of azo-dyes, which are anaerobically degraded by microorganisms into aromatic amines. These substances are confirmed or suspected carcinogens, posing a relevant health risk to humans and other living creatures [9]. Among several pollutant dyes [18], malachite green (MG) is a cationic N-methylated diaminotriphenylmethane colorant used in textile, leather, and paper factories [19]. MG is considered one of the most toxic dyes especially due to irreversible damage to aquatic organisms [19,20,21].

In the best scenario, dye degradation is thermodynamically favored, but kinetically too slow [22,23,24]. Even in the case that the degradation is also thermodynamically unfavorable, a photocatalyst can be used [25]. Photocatalysts are materials capable of absorbing photon energy and “storing” it in an excited state. However, stress must be put on the fact that the efficiency of each photocatalysis step must be maximized (Appendix A) [26]. In particular, photogenerated charge recombination is a typical drawback of heterogeneous photocatalysis systems. Various ways to limit the phenomenon have been studied, such as doping and creation of composites by coupling two or more materials. When a semiconductor is doped, small quantities of a new material are added, so that the band gap energy is modified, and a better performance is achievable. Similarly, a higher number of photocatalysts with distinct band gap energies can be coupled [27,28,29,30,31]. A simple and effective strategy to limit photogenerated charge recombination is to produce nanosized particles of the desired photocatalyst. In doing so, the average length of the photogenerated charge path, from the bulk to the surface reaction site, is greatly reduced. Consequently, the probability of recombination is significantly lower [32]. The use of semiconductor photocatalysts as composites of a wide range of materials for environmental remediation purposes has been extensively reported [33,34,35].

Perovskites are a class of mixed metal oxides, named after the naturally occurring mineral perovskite, CaTiO_3_. Their structure can be schematized as corner-sharing octahedra, oriented along the same axis (Figure 1A). In contrast to the pure perovskite structure, in pseudo-perovskite materials the disposition of the octahedra is distorted. In the last years, Aurivillius phase oxides, originally discovered in 1949 [36], have gained much interest [37,38]. These are layered metal oxides, generally formulated as Bi_2_A_m−1_B_m_O_3m+3_ where A and B are transition metals, O oxygen and *m* a positive integer whose value depends on the specific elements involved and the phase structure. *m* essentially represents the number of consecutive perovskite sheets between two bismuth oxide layers. The Aurivillius structure is formed by (Bi_2_O_2_)^2+^ sheets swapped with one or more pseudo-perovskite layers with formula (A_m−1_B_m_O_3m+1_)^2−^. B can be Fe, Cr, Ti, Ga, Nb, V, Mo, W etc., [39]. In this work we apply Mo and W (separately) as component B (Figure 1A) with condition *m* = 1 such that component A is not present. Aurivillius phases are generally known as oxides in literature [38,39,40,41] and commonly referred to as compounds containing bismuth as a cation, even though they are covalent solids (in the same way that bismuth vanadate, bismuth tungstate are covalent solids) [42]. Bismuth-containing photocatalysts are recurring in literature as standalone or base materials for composite photocatalysts [28,43]. They typically have band gaps in the energy range of visible light. Bismuth oxyhalides (BiOX, with X = halogen) for example, permit a high photogenerated charge mobility due to their electronic structure. The layered structure is very effective in boosting the photogenerated charge separation [43]. BiOX-based composites obtained by solvo-hydrothermal and co-precipitation synthesis methods often reached up to 99% of pollutant degradation efficiency in limited time frame (1–4 h). However, due to the weak strength of the halogen bond [44], bismuth oxyhalides are fairly sensitive (except BiOBr) to photoinduced corrosion.

Aurivillius phases are ferroelectric, that is, they have a permanent electric polarization between the layers, thanks to their particular structure [38,45]. Alternate layers in these materials behave as a two-dimensional capacitor. When *m =* 1 (only one perovskite layer between bismuth oxide layers), the polarization is maximal, greatly improving charge separation. Moreover, narrow band gaps of certain Aurivillius phases allow them to absorb visible light, while remaining generally stable against photocorrosion and leaching of metals in aqueous solution [46]. Bismuth tungstate Bi_2_WO_6_ (BWO) and its isostructural bismuth molybdate Bi_2_MoO_6_ (BMO) both exhibit high ionic conductivity and high polarization (having *m* = 1) [25]. For these reasons, layered bismuth molybdate and tungstate have received much attention recently, and they have been used to photocatalyze dye degradation [40,47,48].

Di and coworkers focused on the importance of oxygen defects on photocatalytic water splitting on BWO nanosheets [49]. Chankhanittha, et al. used BMO for photocatalyzed degradation of two azo dyes and two antibiotics, demonstrating the capability of these solids for removing pollutants from aqueous solutions [48]. Recently, Yin et al. constructed a nanocomposite of BWO and a metal organic framework with the aim of improving charge separation and effectively degrade rhodamine B in aqueous solution through visible light irradiation [50]. Ruan et al. prepared oxygen-vacancy rich BWO microspheres and used them to degrade six types of phenolic compounds [42]. Dutta et al. studied the influence of transition metal dopants on BMO nanorods, used as photocatalysts for methylene blue and MG degradation, reaching up to 32% MG removal with undoped bismuth molybdate [30].

Photocatalytic efficiency of layered bismuth molybdate and tungstate can be improved with the use of composite materials (heterojunction or Z-scheme) [51,52,53,54] or doping [31]. Although several doped or composite photocatalysts such as have been used for dyes and drugs degradation, their preparation requires a complex (time consuming and expensive) multiple-steps synthesis. A wide variety of works reported the use of composites for the degradation of pollutants, however their improvement in physico chemical properties and photocatalytic efficiency is rarely determined by an additional synthetic step only.

In this work, our aim is to prepare highly efficient photocatalysts using a simple synthetic route without the need of composites or the use of expensive reagents. To this purpose, basic syntheses were carried out with just one extra hydrothermal step added to a simple method from literature [45,53]. We synthesized BWO and BMO nanosheets and characterized them through X-ray diffraction (XRD), Fourier transform infrared spectroscopy (FTIR), transmission electron microscopies (TEM), thermogravimetric analysis (TGA) and light scattering (DLS/ELS). We compare the photocatalytic activity of BWO and BMO, examining the photocatalytic degradation of the polluting dye malachite green. Finally, kinetic behavior of BWO and BMO together with reuse stability was determined.

## 2. Materials and Methods

### 2.1. Chemicals

Bismuth (III) nitrate pentahydrate Bi(NO_3_)_3_·5H_2_O ≥ 99.99%, potassium bromide KBr FTIR grade, ≥ 99% purity, polyvinylpyrrolidone (PVP, [C_6_H_9_NO]_n_) (average molar mass of 40,000 g/mol), D-mannitol (C_6_H_14_O_6_) ≥ 98%, sodium molybdate dihydrate (Na_2_MoO_6_·2H_2_O), and sodium tungstate dihydrate (Na_2_WO_6_·2H_2_O) ACS reagents ≥ 99% purity were purchased from Sigma-Aldrich (Milan, Italy). Bi-distilled MilliQ^®^ water was used as solvent for all reactions throughout the whole work and referred to simply as “water” for convenience. The same water was also used for any washing or dilution procedures. Ethanol (for washings only) >96% was purchased from Honeywell (Seelze, Germany). All reagents were used without further purification.

### 2.2. Synthesis of BiOBr Nanosheets

BiOBr nanosheets were synthesized according to the procedure of Di et al. [49,56]. Briefly, a mass of 2.402 g of mannitol was dissolved in 132 mL of milliQ water under magnetic stirring at room temperature. Then, 2.015 g of Bi_2_ (NO_3_)_3_∙5H_2_O and 1.760 g of polyvinylpyrrolidone (PVP) were added to the solution. After they were completely dissolved, 44 mL of aqueous KBr (0.1 M) solution was injected into the first one, over the course of a minimum of 2 min under stirring, which was maintained for further 30 min. The liquid was poured into a 220 mL Teflon-lined stainless-steel autoclave, which was closed and heated in an oven at 160 °C for 3 h and subsequently cooled to room temperature. The nanosheets (NS) were gathered by centrifugation (4500 rpm for 40 min) and washed three times with water and three times with ethanol. After the last separation, the material was left in a 60 °C drying oven overnight, finely ground with mortar and pestle and collected to be used in the following reactions. No buffer solutions or any other pH control strategy was used in the syntheses or the subsequent characterizations and tests.

### 2.3. Synthesis of BWO Nanosheets

A mass of 0.244 g of previously prepared BiOBr NSs (0.244 g) was dispersed in 64 mL of distilled water and sonicated for 40 min and kept under magnetic stirring. Then, 0.528 g of Na_2_WO_4_∙2H_2_O was slowly added to the dispersion, which was stirred for further 30 min. The mixture was then heated in an 80 mL Teflon-lined stainless-steel autoclave at 140 °C for 1 h [49]. The separation, washing, and drying procedures were identical to those used in the BiOBr synthesis.

### 2.4. Synthesis of BMO Nanosheets

In the case of BMO, various synthetic methods were used (Appendix A). The first sample (BMO-0) corresponds to a synthetic strategy already reported in literature [56]. In a typical non-hydrothermal synthesis of bismuth molybdate, a dispersion of BiOBr NSs (0.305 g) in 40 mL of water was prepared by sonication for 40 min. Then, 0.484 g of Na_2_MoO_4_·2H_2_O was added under stirring. The mixture was heated at 50 °C and kept under stirring at this temperature and atmospheric pressure for 24 h. After centrifugation, collection, and re-dispersion of a yellow solid by sonication, another 0.968 g of Na_2_MoO_4_·2H_2_O was added, and the reaction was carried out for 24 h, at 50 °C, under stirring.

The newly developed synthetic method used here involved using 0.457 g of previously prepared BiOBr NS, which was dispersed in 60 mL of distilled water under sonication for 40 min. About 0.726 g of Na_2_MoO_4_∙2H_2_O was slowly added and kept under stirring for 30 min at room temperature. Then the solution was transferred in an 80 mL autoclave and heated at 140 °C in the oven. To investigate the influence of hydrothermal synthesis duration on the properties of the material, sample BMO-3 was heated for 3 h, while sample BMO-24 (throughout the paper BMO-24 is reported as BMO for clarity) spent 24 h in the oven. In both cases, separation, washing, and drying procedures were the same of BiOBr and BWO syntheses. The hydrothermal treatment and the overall shorter time of ion exchange in our synthesis were the key differences compared to literature (Table 1) [49,56].

### 2.5. Samples Characterizations

X-ray diffraction (XRD), transmission electron microscopy (TEM), thermogravimetric analysis (TGA), diffuse reflectance spectroscopy (DRS), dynamic light scattering (DLS), and electrophoretic light scattering (ELS) were used to characterize the three materials. Samples were finally tested in terms of capability of degradation of a dye pollutant in aqueous solution by means of UV-Vis spectroscopy. XRD analysis was carried out through a Bruker D8 Advance equipped with a PSD LynxEye detector. TEM images were acquired on a JEOL JEM 1400 Plus. TGA analysis was made on a Perkin-Elmer Thermogravimetric Analyzer TGA/DSC, DRS was performed on an Agilent Technologies Cary 5000 UV-Vis-NIR spectrophotometer with DRS accessory. FTIR spectra were recorded with a Brüker Tensor 27 FTIR spectrometer equipped with a diamond-ATR accessory. DLS and ELS were both executed by means of a Zetasizer Nano series by Malvern. For the photocatalytic tests, a Cary 60 UV/Vis spectrophotometer by Agilent Technologies was used.

### 2.6. Photocatalytic Tests

All the samples for photocatalytic tests were held in glass vials, gas-tight sealed with screw caps as soon as possible after their preparation. Rubber gaskets were used when necessary. In both BWO and BMO tests, four identical 4 mL glass vials were filled with 4 mL of a 10.4 mg/L solution of malachite green oxalate (MG), and labeled as **1**, **2**, **3**, and **4** (Appendix A). About 4 mg of BWO (BMO) was added both in **1** and **2**. Samples **1** and **3** were sonicated for 20 min, then irradiated for 8 h by a 100 W, 6500 K LED floodlight, under constant magnetic stirring. Samples **2** and **4** were kept in the dark for the same amount of time. A sample of the 10.4 mg/L MG solution, as well as samples from all vials after the irradiation, was collected and characterized via UV-Vis spectroscopy immediately before and after the experiment. Centrifugation was used to separate liquid from solid in cases **1** and **2** after the experiment. The effective concentration of MG (Appendix A), as well as the effective mass of BWO or BMO, was recorded and reported in the results section, to precisely compare the effects. MG removal percentage was calculated as in Table 2 (Appendix A), where *A*_0_ is the absorbance of the initial 6.9 mg/L MG solution and *A* the absorbance of the proper sample.

### 2.7. Photocatalysts Reuse

The reuse of BWO and BMO photocatalysts was carried out in transparent centrifuge plastic tubes with sealing screw caps. Each tube was filled with 10 mL of MG aqueous solution (conc. 6.97 mg/L), 10 mg of BWO or BMO, and a small stirring bar. The photocatalysts were dispersed by ultrasonication for 20 min, then were positioned at the same distance in front of the lamp. Two identical stirring plates were used, and the position of each photocatalyst was switched at every reuse cycle. After 24 h of irradiation, the tubes were centrifuged at 4500 rpm for 15 min, leading to complete precipitation of the solid. The supernatant was quantitatively removed and immediately analyzed with UV-Vis spectrophotometry. The tubes were then filled again with MG solution, ultrasonicated and put back in front of the lamp for a new reaction cycle.

## 3. Results and Discussion

### 3.1. Physico-Chemical Characterizations

BiOBr was initially synthesized and then used as a hard template for the preparation of the two photocatalysts BWO and BMO (Figure 1B). Three different BMO samples were synthesized, the first following a procedure from literature (BMO-0) [49], and other two involving an additional hydrothermal step in an autoclave at 140 °C for 3 h (BMO-3) and 24 h (BMO), respectively (Figure 1B).

#### 3.1.1. XRD and TEM Analysis

XRD patterns and TEM images of BiOBr, BWO, and BMO samples are shown in Figure 1. The diffractograms are consistent with those reported in the literature [49,56], and were reproducible across the different batches of synthesized materials. BMO samples show a clear increase in peak sharpness and resolution as a result of the hydrothermal treatment (Appendix A), reaching the best resolved XRD pattern in the sample kept for 24 h in autoclave (BMO). Table 1 reports the mean dimensions of crystallites (<D_XRD_>) calculated through the Scherrer equation [57] with a Warren correction [58]. Appendix A highlights the positions of the peaks used for crystallite size calculation through Scherrer theory. Because many peaks tend to overlap, these positions can change for the different materials. BWO nanosheets showed a mean lateral dimension of 11 nm which was comparable with that of the BiOBr precursor (14 nm) (Appendix A). The mean crystallite size of BMO samples increased due to the hydrothermal treatment, being 11 nm, 25 nm, and 60 nm for BMO-0, BMO-3, and BMO, respectively. Crystallites size determined by Scherrer method was compared with Williamson–Hall (W-H) analysis [59].

The following form of the W-H equation was used:(1)$$\beta \cdot cos\left(\theta \right)=\left(\frac{k\cdot \lambda }{d}\right)+\left(\eta \cdot sin\left(\theta \right)\right)$$
where β is the full width at half height of the peak, d the average crystallite size (nm), λ the wavelength of Cu kα radiation (1.54056 Å), θ the Bragg angle (degrees), and k is a dimensionless shape factor, typically assumed to be 0.9 [60]. η considers the internal strain broadening. Notably, when η=0, the Scherrer equation is obtained. The experimental XRD peaks were fitted to obtain β. Angle was expressed in θ degrees. cos(θ) and sin(θ) were calculated, and β·cos(θ) was plotted against sin(θ). A lattice dilatation (η > 0) is present for BWO and BMO (Figure 2) while BiOBr shows a compressive strain (η < 0) and a poor adherence to liner fit (Appendix A). d values were slightly lower than those calculated with Scherrer equation (25 and 48 nm for BWO and BMO respectively). d was not computable for BiOBr. BMO shows the best convergence to a linear fit (Figure 2B) suggesting a higher crystallinity than BWO likely due to longer time in the autoclave.

**Table 1 ijms-23-05422-t001:** Mean dimensions of crystallites (<DXRD>) are calculated from full width at half maximum (FWHM) of X-ray diffraction peaks a of BiOBr and BWO.

Sample	Autoclave Time (h)	D_XRD_ (nm) ^a^
BiOBr	3	14
BWO	1	11
BMO-0	0	11
BMO-3	3	25
BMO	24	60

^a^ Scherrer equation: $D_{XRD}=\left(K\cdot \lambda \right)/\left(FWHM\cdot cos\left(\theta \right)\right)$. Pure Al_2_O_3_ powder is used as external standard. Warren correction [58] through the external standard method, $FWHM_{sample}=\sqrt{FWHM_{exp}{}^{2}+FWHM_{std}{}^{2}}$ where, FWHM is full width at half maximum with subscripts indicating sample signal (SAMPLE), experimentally obtained signal (EXP), and external standard signal (STD), respectively.

A TEM image of the BiOBr precursor (Figure 1D) shows nanosheets (NS) having irregular shapes which are almost transparent to the electron beam, suggesting an atomic-scale thickness. The lateral dimension of BiOBr NS is around 50–80 nm in agreement with the literature, although an even wider range, between 30 and 160 nm, is also reported [56]. The morphology of BWO NS was like that of BiOBr with a very tight range of dimensions (between 40 and 60 nm, Figure 1E). The BMO sample showed a sharper square morphology with NS generally of larger size than BiOBr and BWO samples (Figure 1F).

#### 3.1.2. FTIR Spectroscopy

The FTIR spectrum of the BiOBr precursor (Figure 2A) displays characteristic bands at 536 cm^−1^, 691 cm^−1^ corresponding to Bi-O stretching vibrations, 830 cm^−1^ (typical of tetrahedral MO_4_ vibrations as in VO_4_), 1278 cm^−1^, and 1421 cm^−1^ corresponding to Bi-Br [61]. Figure 2A also shows the FTIR spectra of BWO and BMO. Signals in the 400–1000 cm^−1^ region are attributed to the pseudo-perovskite layer of the Aurivillius phases (Mo-O stretching, Mo-O-Mo bridging stretching vibrations [62]). In particular, the peak at 691 cm^−1^ is due to the asymmetric vibration of apical oxygens in MoO_6_ octahedra in the BMO sample [62]. The peaks at 830 cm^−1^ and 1421 cm^−1^ can be related to W-O stretching and W-O-W bridging vibrations in BWO [63]. The disappearance of Bi-Br bands at 1278 cm^−1^ and 1421 cm^−1^ in both the BWO and BMO spectra demonstrates that the ion exchange process, in favor of WO_4_ or MoO_4_, took place. On the contrary, bands around 536 cm^−1^ and 691 cm^−1^, after the reaction on BiOBr were more intense, especially for BMO.

#### 3.1.3. TGA

Thermogravimetric analysis (TGA) of BiOBr, BWO, and BMO samples was then carried out (Figure 3B). Three main mass losses occur in the BiOBr TGA and DSC plots as detailed in Appendix A. BiOBr mass losses are 15%, 3.5%, and 17.5% in the temperature range of 240–430 °C, 430–520 °C, and 520–850 °C, respectively. The first loss can be attributed to decomposition of bismuth oxybromide into bismuth oxide and bromine. Decomposition of residual bismuth nitrate, forming NO_2_ and O_2_, is attributable to the second weight loss. The third mass loss at T > 520 °C is likely due to decomposition and phase transition of the bismuth oxides. A very small mass loss (2.5%) in the range 240–430 °C is observed for BWO/BMO. No further decomposition or phase change takes place for T > 430 °C.

In the case of BMO, an analogous small weight loss of 2.5% is observed in the same temperature range. Interestingly, the other two samples with no (BMO-0) or a short (3 h in BMO-3) hydrothermal treatment showed higher mass losses (8% and 7%, respectively) in the range 240–700 °C (Appendix A), suggesting a higher degree of impurities likely due to residues of either BiOBr or Bi(NO_3_)_3_ (Appendix A) [64,65].

#### 3.1.4. DLS and ELS Analysis

DLS and ELS measurements of aqueous dispersions of BiOBr, BMO, and BWO NSs are shown in Figure 3C. DLS and ELS measurements for BMO-0 and BMO-3 are shown in Appendix A. No buffer or salts were added. All dispersions were around 2 mg/L of concentration, had a pH of approximately 6.8 ± 0.3, and were stable for about 2–3 h. The BiOBr precursor NS had a hydrodynamic size like those of BWO. More precisely, BWO NSs are slightly smaller (75 ± 2 nm) than those of BiOBr (91 ± 5 nm). Zeta potentials are low in absolute value (+3 mV for BiOBr and +7 mV for BWO), leading to unstable colloidal dispersions. BMO showed the highest hydrodynamic radius (145 ± 28 nm) with a negative zeta potential (−11 mV). According to the Stokes–Einstein equation, the hydrodynamic size is the diameter of a hypothetical rigid sphere which diffuses with the same diffusion coefficient of actual samples. Hence, for geometrical reasons (they are nanosheets and not spheres), it is only an indicative size of BiOBr, BWO, and BMO samples once dispersed in water.

#### 3.1.5. DRS Analysis

Figure 3D and Appendix A show the results of diffuse reflectance spectroscopy (DRS). BiOBr and BMO have an absorbance peak at about 355 nm, although the former shows non zero absorbance in the visible region [66,67,68].

The absorption peak of BWO is blue-shifted to 312 nm, that is shifted deeper into the mid-UV. Less visible light is absorbed, as the tail rapidly decays, reaching almost zero at 480 nm [69].The presence of negative absorbance values can be linked to the intrinsically low repeatability of powdery sample positioning inside the sample holder of the instrument. The global interpretation of DRS data requires caution. When considering the complete photocatalytic system (e.g., the photocatalyst dispersed in an aqueous dye solution), absorption can either shift further into the UV or more toward the visible region of the electromagnetic spectrum. Indeed, complex phenomena of sensitization and energy transfer can, in principle, take place [29]. The absorption edge wavelength for BWO was estimated to be 440 nm, and 527 nm for BMO (Figure 3D). To better determine the band gap potential (E_g_) of our materials, Tauc plots for BMO and BWO were created, by plotting ${\left(\alpha \cdot h\cdot \nu \right)}^{2}$ versus h·ν in eV, which represents the photon energy (Appendix A) where α represents here the absorption coefficient of the material, h is Planck’s constant and ν is the frequency of the radiation [70,71]. The equation assumes a direct allowed electronic transition. Fitting the linear part of the Tauc plot with a straight line, the intercept on the *x*-axis represents the band gap energy E_g_. BWO gave a calculated E_g_ of 3.26 eV, BMO had E_g_ of 3.27 eV [72].

The electronegativity of elements (Bi, Mo, W, O) contained in our photocatalysts was calculated by arithmetical averaging between their electron affinity and energy of first ionization (in eV). These electronegativity values, calculated as per Mulliken theory [73], were then used to calculate the electronegativity of BWO and BMO. These values were obtained by geometrical averaging between the constituent elements, resulting in 6.21 eV and 6.13 eV for BWO and BMO, respectively. Then, Equation (2) was applied.
(2)$$E_{VB}=X-E_{e}+\frac{E_{g}}{2}$$
where *X* represents the Mulliken electronegativity, calculated as explained above; *E_VB_* is the potential of the valence band, *E_e_* is the absolute potential of the hydrogen standard electrode (SHE), which is estimated to be 4.2 eV [74], and E_g_ represents band gap in eV. The calculation gave 3.34 eV and 3.26 eV of *E_VB_* for BWO and BMO respectively. The difference between the valence band potential and the band gap permits estimation of *E_CB_*, conduction band potential (in eV), as in Equation (3). The conduction band potentials E_CB_ obtained were 0.08 eV and −0.01 eV for BWO and BMO.
(3)$$E_{CB}=E_{VB}-E_{g}$$

Appendix A shows two hypothetical schematics for photocatalytic production of the hydroxyl radical.

E_0_ = 2.40 eV is the electrochemical potential for the redox couple H_2_O/OH, and E_0_ = −0.18 eV for the O_2_/O_2-_ couple, which is related to the electrochemical synthesis of superoxide anion [75]. Because the potential of VB is more positive than E_0_(H_2_O/OH), the hydroxyl radical is produced in the system (formally, water reacts with a hole h+). The potential of the CB should be more negative than E_0_(O_2_/O_2_), but this is not the case, as BWO has E_0_(CB) = 0.08 eV and BMO has E_0_(CB) = −0.01 eV. This indicates that superoxide is likely not produced in appreciable amounts. The hydroxyl radicals produced upon irradiation are well-known to cause MG degradation, representing in fact a common reaction pathway. Depending on the specific degradation products, MG could react with the electrons accumulated in the CB, as already previously found [76,77].

### 3.2. Photocatalytic Removal of Malachite Green from Aqueous Solutions

Photocatalysts need to be considered in relation with their real applications [78,79]. To test the above characterized samples for environmental remediation, photocatalytic tests on BMO and BWO were carried out to evaluate the degradation of an organic pollutant, the dye malachite green (MG). UV-Vis spectroscopy was used to quantify the photocatalyzed degradation of MG, calculated by means of Appendix A [80].

#### 3.2.1. Kinetic Analysis

A kinetic study with BWO and BMO was performed to determine MG concentration in solution as a function of time (Figure 4). The points in Figure 4A,B were fitted with an arbitrary curve as visual guidance. It can be seen how BWO has a different behavior compared to BMO. The former starts with a fast removal rate, with a MG removal up to 89% after 24 h. The latter, even with a slower removal rate, reaches slightly higher overall percentage (91%) at 24 h. Both photocatalysts achieved maximum MG degradation after 24 h. BWO and BMO spectra (Figure 4C,D) show a decrease of the absorbance at 617 nm together with a peak shift toward lower wavelengths.

The blue-shift of the peak at 617 nm is particularly evident for BMO samples (Figure 4D) and suggests cleavage of the chromophore. BWO showed a bigger area under the spectra at low wavelengths (200–400 nm) when compared with BMO (well visible after 25 h of irradiation) proving a higher degree of MG degradation in the solution containing BMO [76]. MG degradation generally involves demethylation and breakdown by generating degradation products such as benzophenone derivatives [17,22,76,81]. Because of health and environmental hazards caused by chemicals such as benzophenone derivatives [82], a high degree of decomposition is always desirable. Langmuir–Hinshelwood model usually describes the kinetic expression of a two-phase heterogeneous catalytic process at the catalyst surface [83,84,85,86,87] according to Equation (4):
(4)$$r=-\frac{k_{0}KC}{1+KC}$$
where *C* is the compound concentration, *k*_0_ the reaction rate constant, and *K* the adsorption coefficient of the organic compound on the catalyst. When *KC* is <<1, Equation (4) can be simplified to a pseudo-first order kinetics model:(5)$$r=-k_{0}KC=-kC$$
where *k* includes *k_0_**K*. Kinetic data of MG for BMO and BWO were fitted according to the integrated form (with *C* = *C*_0_ at *t* = 0 and *C* = *C* at *t* = *t*) of Equation (5) (Appendix A):(6)$$ln\left(\frac{C}{C_{0}}\right)=-k\cdot t$$
where, *c*_0_ is the initial concentration of MG (mg/mL), *c* is the concentration of MG (mg/mL) at irradiation time *t* (min), and *k* is the kinetic constant (min^−1^). The kinetic constants for BWO and BMO, calculated with Equation (6), are listed in Appendix A. BMO kinetic data showed a better fit (R^2^ = 0.95) for the model than those of BWO (R^2^ = 0.75). This result suggests the occurrence of a typical heterogeneous photocatalytic process for BMO [88].

#### 3.2.2. Photocatalytic Tests

To understand MG degradation mechanism by BWO and BMO, controlled photocatalytic tests were performed. Photocatalytic experiments were designed to fully consider all the possible phenomena that could lead to a decrease of MG concentration and to distinguish photocatalytic effects from adsorption effects. Due to the quite high value of MG dye molar extinction coefficient (ε_(MG)_ = 1.33 × 10^5^ L mol^−1^cm^−1^), concentration was kept relatively low (6.97 mg/L) to avoid light penetration issues into the bulk solution [89,90,91,92]. Briefly, the MG aqueous solution was put in four different vials. In two of them BWO photocatalyst was added (samples **1** and **2** in Figure 5A), while no photocatalyst was added in the remaining two vials (samples **3** and **4**). Samples **1** and **3** (Figure 5A) were then irradiated with visible light for 8 h, while samples **2** and **4** were kept for the same time in the dark. Sample **1**, containing the dispersion of BWO in the MG solution and irradiated with visible light for 24 h, was almost fully decolored. In particular, 89 ± 5% removal of MG was obtained (Figure 5B). Sample **2**, which was kept in the dark, resulted in a 51 ± 0.1% decrease of MG concentration (Figure 5A,B). This decrease is likely due to adsorption on the BWO surface rather than photocatalysis [77,93]. Moreover, sample **3** in the absence of photocatalyst shows that the uncatalyzed photodegradation of MG contributes approximately 31 ± 0.2% (Figure 5B). Finally, in the absence of both photocatalyst and light (sample **4**), the aqueous dye solution resulted in a 5 ± 0.1% decrease of absorbance (Figure 5B), which suggests the spontaneous oxidation of MG. MG is known to degrade in artificial light, and even more under sunlight [22,77]. The contribution of photolysis (which takes place in sample 3) has a non-negligible effect compared to the control (sample **4**). In sample **2**, although dye adsorption on the NSs can be an alternative method for pollutant removal, it could also be a needed preliminary step before the photocatalytic process takes place. The higher MG removal in sample 1 compared to sample **2** is in agreement with what is reported in literature by Chen et al. [93]. In their work they found a substantial MG removal in the presence of the photocatalyst and light. In the absence of light, BWO adsorbs MG without degrading it. The simultaneous occurrence of adsorption and photocatalysis has been previously reported in various photocatalytic materials such as CeO_2_/CdO, guar gum/Al_2_O_3_, and Cu/ZnO [90,94,95]. The UV-Vis spectra of all the solutions involved are shown in Figure 5A, and in particular the shift of the main absorption peak of sample 1 from 617 nm to 587 nm is indicative of the transformation of the chromophore into a smaller resonant system. Possible mechanisms of MG degradation in various photocatalyzed and even bio-photocatalyzed systems have been investigated in literature [17,77,90].

A similar experimental set-up was followed for BMO photocatalyst (Figure 5B,C). Sample **3** and **4** were in common with the BWO experiment. Under 24 h visible light irradiation, BMO resulted in a MG removal of 91 ± 2%. Remarkably, Dutta et al. [30] who studied MG degradation with doped and undoped BMO nanorods photocatalysts, achieved only a 32% of MG removal with the undoped photocatalyst. In the absence of light, the same BMO sample could achieve up only to 19 ± 0.1% removal, indicating a less predominant contribution from adsorption to the overall degradation process. Analogously to BWO, for BMO (sample 5) it is also possible to observe a shift in the main absorption peak, from 617 nm to 593 nm. Again, this is considered indicative of the dye degradation [17,77,90]. BMO (with 24 h hydrothermal treatment) showed the best removal performance due to photocatalytic process achieving the highest dye degradation percentage, even with a reduced adsorption contribution. Similarly Saad et al. [89] used chitosan/Ce–ZnO composites synthesized under microwave irradiation and they achieved up to total MG degradation with a 20% of adsorption contribution. In another photocatalytic composite based on doped Co-ZnO/algae study, almost 99% of MG degradation was reached but with a very high adsorption contribution of 50% [96]. It is noteworthy that BMO-0 (no hydrothermal step) gave different photocatalytic results compared to BMO, although a high (85%) removal of MG was obtained, this is likely due to adsorption (79%) rather than photocatalysis (Table 2). The great difference in adsorption contribution between BWO and BMO can be linked to the duration of the hydrothermal steps. A 24-h high temperature residence time led to a higher crystallinity and less defectivity, which agrees with XRD data (Figure 1, Table 1). The decrease in surface defects could explain the dye’s lower affinity for BMO.

#### 3.2.3. Reuse Measurements

To evaluate the stability of BWO and BMO over multiple cycles, a 24-h reuse experiment was performed (Figure 6). Both BWO and BMO showed extremely good recyclability and no appreciable activity loss with a remarkable MG removal percentage (up to 95%) up to 10 cycles (Figure 6A) was found. Spectra of the starting MG solution and the two supernatants of BWO and BMO photocatalysts after 24 h at cycle 8 are shown in Figure 6B. In contrast to BWO, BMO clearly showed a higher degree of MG degradation with almost no absorbance even at low wavelengths (200–400 nm) as seen in Figure 6B, consistent with kinetic spectra in Figure 4C,D. Good reuse performance for MG degradation was shown by Magdalane et al. with CeO_2_/CdO multilayered nanoplatelet photocatalysts. In their work they achieved up to three cycles with only a 5% activity loss when compared with the first use [90]. Reuse efficiency in MG degradation was described also by Saad et al. with chitosan-supported ZnO and Ce–ZnO nano-flowers. In their work they registered a 10% activity loss after four cycles and 16% loss after five cycles [89]. The good reuse performance, together with the high decomposition degree, emphasizes the possible use of both photocatalysts and especially of BMO for MG degradation. Compared to analogous procedures in literature, BMO NSs were synthesized from cheap reagents without the additional use of composites, maintaining a high photocatalytic performance [49,56]. The photocatalytic setup demonstrated a very interesting difference between BWO and BMO. While for BWO both photocatalysis and adsorption phenomena are at work, for BMO, dye adsorption contributes to a lower extent to the degradation process. This makes it possible to avoid a great deal of contamination, achieving excellent results as a photocatalyst rather than as an adsorbent.

## 4. Conclusions

Two Aurivillius oxide photocatalysts, bismuth molybdate (BMO) and bismuth tungstate (BWO) nanosheets, were synthesized starting from bismuth oxybromide (BiOBr) as the hard template [56]. All three materials were characterized to completely determine the structural and chemical features. XRD analysis showed a better resolution and sharper peaks of BMO when compared with BWO and BiOBr. Crystallite sizes estimated by XRD were around 11 nm for BiOBr and BWO, with a significant increase for BMO (up to 60 nm) likely due to the hydrothermal step. TGA analysis showed a small mass loss of 2.5% at 240–430 °C for BWO and BMO indicating a negligible presence of residual BiOBr in both samples. BMO-0 and BMO-3 with no and short hydrothermal treatment, instead, showed 7% and 8% of mass loss respectively indicating a residual contamination due to either BiOBr or Bi(NO_3_)_3_. After physico-chemical characterization, BWO and BMO nanosheets were tested for the photocatalyzed degradation of malachite green dye. MG removal kinetics data were fitted with the Langmuir—Hinshelwood model indicating the presence of photocatalytic reaction mechanism for BMO. BWO reached up to 89% of MG removal under light irradiation with a substantial contribution of surface adsorption (51%). On the other hand, BMO achieved 91% of MG removal with only 19% due to adsorption in absence of light indicating the fundamental contribution of photocatalysis rather than adsorption to the removal process. The limited occurrence of dye adsorption is likely an advantage for a photocatalyst since it could result in a surface contamination that may negatively affect the photocatalytic process. Reuse tests of photocatalysts were studied indicating very low efficiency loss up to 10 cycles. In kinetic studies and photocatalytic tests, BMO showed a higher decomposition degree than BWO, suggesting a better performance for the total degradation of organic dyes likely due to the high crystallinity degree. The new synthetic process developed for bismuth molybdate allowed obtaining a pure and efficient photocatalyst with simple synthesis and cheap reagents. Further research is desirable to determine the exact mechanisms of the dye degradation reaction and the influence of synthesis conditions such as the hydrothermal step on the photocatalyst performance.

## Data Availability

The data presented in this study are available in the article and in Appendix A.

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
