# Peer review of "Aurivillius Oxides Nanosheets-Based Photocatalysts for Efficient Oxidation of Malachite Green Dye"

_ijms, 2022, doi:10.3390/ijms23105422_

Round 1

Reviewer 1 Report

Dear . Jon Gao

Assistant Editor: International Journal of Molecular Sciences

Thank you for your attention. I revised the manuscript which referenced as: ijms-1682923 Title: Aurivillius Oxides Nanosheets based photocatalysts for efficient oxidation of malachite green dye. I severely agree a work to be useful for young researchers, especially, to encourage them and gain their information without increasing the length. Hence, good literature review is very important. High data and results obtained with sufficient characterization. The results are good for publication in your journal, because of the novelty of the work, after a moderate revision considering the following comments.

  1. Abstract should be more quantitative by adding more findings of the work.

  1. Introduction section, A: Should be divided to separate paragraphs depending on the mater discussed. B) The first paragraph should be enhanced by mentioning various pollutants such as heavy metals, organic/inorganic pollutants etc and the importance of both removal and quantification. Please read. J. Electroanal. Chem. 810 (2018) 119-128 and 829 (2018) 95-105.

Then bold dye pollution, I suggest read. Spectrochim. Acta Part A: Molecul. Biomolecul. Spect. 196 (2018) 334–343

  1. C) Please bold Bi-contained semiconductors in photocatalysis, please read Photochem. Photobiol. A: Chem. 389 (2020) 112223, J. Solid State Chem. 310 (2022) 123018.
  2. D) E/h recombination drawback of heterogeneous photocatalysis should be mentioned briefly with various strategies used for diminishing it, plz read J. Hydrogen Energy. 45 (2020) 24636-24656. Then bold how it diminished in this work.

  1. Please check the experimental section and write clearly the used conditions (volumes in mL, pH, concentration etc.) used in performing the experiments. This section should be clearly usable by readers. If procedures used from literature, please cite them.

Two different synthesis 8sed should be clearly described and the difference n that adopted from literature should be mentioned.

In my opinion, experimental section must be as the second part of the work no 3rd one.

  1. Head line of ‘Results’ ‘should be ‘Results and discussion’ section.

  1. Please divide each characterization section as a separate subtitles and enhance the discussion based on the next comments.

  1. In XRD section, I suggest read this work and estimate the crystallite size by W-H methods and compare with that of obtained by Scherrer model. Then conclude is there any strain effect on the size broadening or not? Inter. J. Hydrogen Energy 45 (2020) 33381-33395.

XRD pattern show various peak width for the samples, and this comparison is nice

  1. Detailed discussion on FTIR spectra of Bi-O has been presented in, plz read for extending the discussion. Mater. Res. Bull. 151 (2022) 111830.

  1. Discussion on Kubelka-Munk and Tauc plots is poor, Please read carefully and enhance it and compare the values obtained for the samples based on the synthesis procedures used. Inter. J. Hydrogen Energy 45 (2020) 24749-24764. J. Photochem. Photobiol. A: Chem. 348 (2017) 68–78.

I suggest estimate potential positions of CB and EB levels by reading following works Desal. Water Treat. 166 (2019) 92–104

Then draw suitable Schematic for each system, compare the potential positions with MG, and enhance discussion in various efficiencies of the catalyst in photodegradation section, please read following works and used various potential positions mentioned for superoxide and hydroxyl radicals. Sep. Purification Technol. 235 (2020) 116228, Composites Part B 183 (2020) 107712

  1. From TG data, I suggest do a quantitative calculation based on the mass losses observed. Enhance discussion on the different or more intense weigh loss peaks for the third sample with others.

  1. The Hinshelwood model is a concentration dependent model that well described in Desal. Water Treat. 162 (2019) 290-302, Chem. Phys. Lett. 759 (2020) 137873. Plz read carefully and enhance the discussion.

Draw plots of C/Co and ln C/Co versus time. Compare k-values obtained for the samples and get a suitable conclusion for their efficiency.

  1. Following works for photodegradation of MG are good, plz read and enhance photodegradation results. Internat. J. Photoenergy Volume 2011, Article ID 518153, 10 pages. J. Chem. Vol. 2013 , Article ID 104093, 11 pages, J. Ind. Eng. Chem. 20 (2014) 2719–2726..

  1. In Table 2, plz compare the various surface adsorption efficiency and give some reasons for very lower efficiency by BMO.

  1. Please enhance quality of all figs, all have poor quality. Font size is also poor.

The best regards

Author Response

We thank Reviewer 1 for his/her comments which have contributed to the manuscript improvement. A detailed response is attached in the uploaded response.

Reviewer 2 Report

Comments on Aurivillius Oxides Nanosheets based photocatalysts for efficient oxidation of malachite green dye 

1. The abstract needs to focus on main finding and novelty.
2. Please improve the novelty in the introduction with focus of recent publications in this area. You can use below article:
- https://doi.org/10.1016/j.jhazmat.2021.126986
3. Please add the brand of all equipment.
4. Which peak has been selected for size calculations.
5. There are many typos in throughout of the manuscript.

Author Response

We thank Reviewer 2 for his/her comments which have contributed to the manuscript improvement. A detailed response is attached in the uploaded response.

Author Response

We thank Reviewer 3 for his/her comments which have contributed to the manuscript improvement. A detailed response is attached in the uploaded response.

Round 2

Reviewer 1 Report

Dear . Jon Gao

Assistant Editor: International Journal of Molecular Sciences

I studied revised version of the manuscript which referenced as: ijms-1682923 Title: Aurivillius Oxides Nanosheets based photocatalysts for efficient oxidation of malachite green dye. It was relatively enhanced during the revision, but ref list should be checked and corrected before the acceptantnce.\

Some journals' names are missed in the cited refs, please check all and correct carefully.

For example in ref 6, 'Elsevier B.V, 2018; Vol. 810; ISBN 3153292515 ' should be corrected to ' J. Electroanal. Chem. 810 (2018) 119-128'.

Ref 17 2019, doi:10.2174/1381612825666191021142026.

Ref 28, it seems it is book, please complete information.

Ref 57 2019, 61, 54–59, 702 doi:10.1016/j.nanoen.2019.04.029.

Ref 58 Elsevier, 2014; pp. 3–38.

Ref 70 2020, 030149, 2–6.

Ref 82 2008, 70, 2068–2075, doi:10.1016/j.chemosphere.2007.09.008

Etc.